

# Impact of parents' physical activity on preschool children's physical activity: a cross-sectional study

Chang Xu[1], Minghui Quan[1], Hanbin Zhang[1,2], Chenglin Zhou[1] and PeiJie Chen[1]

[1] School of Kinesiology, Shanghai University of Sport, Shanghai, China
[2] Health Promotion Center, Zhejiang Provincial People's Hospital, Hangzhou, China

## ABSTRACT

**Purpose**. This study examined the associations of physical activity levels between parents and their pre-school children based on gender and weekday/weekend.
**Method**. A total of 247 parent-preschool child triads from Shanghai, China were analyzed. The children had a mean age of $57.5 \pm 5.2$ months. Both sedentary behavior and physical activity were measured in all participants using an ActiGraph GT3X[+] accelerometer over seven consecutive days from Monday through the following Sunday. A multivariate regression model was derived to identify significant relationships between parental and child physical activity according to gender and weekday/weekend.
**Results**. There was a significant correlation between mothers' and girls' moderate-to-vigorous physical activity (MVPA) and total physical activity (TPA) on weekdays. Fathers' MPVA levels correlated significantly with those of boys and girls, with paternal influence appearing to be stronger than maternal influence. However, there was not a significant correlation between fathers' and children's TPA. TPA levels of both mothers and fathers correlated with those of girls, but not with those of boys. Parental sedentary levels on the weekend correlated significantly with girls' levels, but not with boys' levels. Children's physical activity levels on weekends were influenced more by fathers' activity levels than by mothers', while the opposite was observed on weekdays.
**Conclusion**. Sedentary behavior and physical activity levels of parents can strongly influence those of their preschool children, with maternal influence stronger during the weekdays and paternal influence stronger on the weekends. Parents' activity levels influence girls' levels more strongly than they influence boys' levels.

Corresponding authors
Chang Xu, xuchang@sus.edu.cn
PeiJie Chen, chenpeijie@sus.edu.cn

## INTRODUCTION

Prevalence of obesity and overweight continue to increase among preschool age children in China (*Xiao et al., 2015*; *Zhou et al., 2014*). Data from the International Obesity Task Force, World Health Organization, and US Centers for Disease Control and Prevention show that among preschool children with a mean age of 5.02 years in northeast China, the prevalence of overweight is 11–12% and the prevalence of obesity is 6–14% (*Ma et al., 2011*). Data from these three organizations indicate that among preschool children

aged 3–6 years in Shanghai, the prevalence of overweight is 13–17% and the prevalence of obesity is 6–11% (*Quan et al., 2014*).

Inadequate physical activity and excessive sedentary time are the primary causes of overweight and obesity in preschool children (*Dennison, Erb & Jenkins, 2002*; *Epstein et al., 2000*; *Gortmaker et al., 1999*). Increasing preschool children's physical activity, especially the level of moderate-to-vigorous physical activity (MVPA), can improve issues with overweight, obesity, cardiopulmonary function and bone density (*Physical Activity Guidelines Advisory Committee, 2008*; *Strong et al., 2005*). In addition, increasing MVPA can promote cognitive development in preschool children and youth and improve their academic performance (*Fedewa & Ahn, 2011*; *Sibley & Etnier, 2010*). In this way, promoting physical activity may benefit preschool children and youth both physically and mentally.

Environmental factors likely affect the behaviors and habits of preschool children, including their tendency to engage in MVPA (*Stokols, 1992*). These factors can originate at home (*Spurrier et al., 2008*), in the preschool (*Kreichauf et al., 2012*) and within the broader community (*Roemmich et al., 2006*)—with the home environment appearing to exert the strongest influence (*Sterdt et al., 2014*). As the earlist, most direct and lasting campanion, parents will affect preshool children's behavior acquisition for a very long time, even for the whole life. As a result, so as to promote preschool children and adolescents' physical activity, their parents' physical activity styles and levels turn into hot issues to be researched by relevant scholars.

For example, preschool children's physical activity correlates with that of their parents (*Cools et al., 2011*; *Dowda et al., 2011*; *Oliver, Schofield & Schluter, 2010*). Parental MVPA levels show a significant association with total physical activity (TPA) levels in preschool children, although this association weakens as the child gets older. Children with two active parents are 5.8 times more likely to be active than are children with two inactive parents (*Oliver, Schofield & Schluter, 2010*). Parental physical activity influences children's activity directly or indirectly by affecting children's self-efficacy (*Trost et al., 2003*). Parents have been shown to influence their preschool children's physical activity by acting as role models or playmates, emphasizing the importance of physical activity, and setting goals for sports skills (*Cools et al., 2011*).

The relationship between parents' and children's physical activity appears to be complex. The child's gender may play a role, with some work suggesting that parents affect the physical activity of their sons more than the activity of their daughters (*Sterdt et al., 2014*). As found by another research, parental sedentary levels correlated significantly with girls' levels, but not with boys' levels (*Jago et al., 2010*). Under some conditions, there may be no relationship at all between parental and child activity, suggesting that the relationship may vary with environmental conditions and raising the possibility that some preschool children are naturally physically active without parental intervention (*Hesketh, Hinkley & Campbell, 2012*; *Taylor et al., 2009*). On account of the discrepancies of the different research above, further study is still needed about how much parental factors affect preschool children's physical activity action and characteristics.

While adults typically engage in various types of physical activity (work, housework, physical exercise, entertainment), preschool children engage predominantly in entertainment, which involves brief but frequent bursts of activity. Measuring such activity can be challenging. Although direct observation is considered the gold standard for measuring physical activity, it may not be feasible in many situations because of expectancy bias, observation effects, and even issues of privacy for study participants. Self-report instruments such as questionnaires can be used to measure physical activity, but children younger than 10-11 years may not have the necessary cognitive skills to accurately report physical activity levels. For children, then, objective tools such as motion sensors can be the most appropriate method for assessing physical activity levels. Pedometers and actigraphs have been used to measure numbers of steps in youth studies (*Kelder et al., 1994*; *Matusik & Malecka-Tendera, 2011*; *Taylor et al., 2013*), but merely measuring the number of steps can lead to a distorted, inaccurate understanding of physical activity.

A superior alternative to assessing children's physical activity levels may be to use pedometers and accelerometers. With the rapid development of computer science and technology, the application of accelerometer not only reduce the errors of retrospecting physical activity by ones' memories, still this can measure the time needed to achieve a given intensity of activity as well as estimate energy consumption. As a result, accelerometer has been widely used in the measuring of children and adolescents' physical activity (*Hnatiuk, 2014*). The newest-generation accelerometers, such as the ActiGraph GT3X (*Yam et al., 2011*), which can measure accelerations of three directions as frontal axis ($X$ axis, fore-and-aft direction), sagittal axis ($Y$ axis, left and right direction) and vertical axis ($Z$ axis, upward and downward direction), then turn it to electrical signal through inner chip sensing sensor, and then turn electrical signal to counts number and to be outputted, lastly classify the counts number according to intensity division points and obtain the accumulative time of SB, LPA, MPA and VPA. This can provide a more comprehensive understanding of children's physical activity,and then gradually replace the methods of previous,such as International Physical Activity Questionaire (IPAQ), Children's Leisure Activities Study Survey (CLASS), motion sensors and pedometers.

Based on the existing relevant researches and circumstances of Chinese preschool children physical activity, this research will adopt a cross-sectional study design, to probe into the relevance between parental physical activity level between preschool children activity level. Supposing that parents' physical activity levels have great effect on preschool children physical activity level, and the effect may exist differences for the gender and time range (weekday, weekends) factors.The ActiGraph GT3X$^{+}$ (Actigraph LLC, Pensacola, FL, USA) accelerometer was used to measure both sedentary behavior and physical activity of preschool children and their parents living in Shanghai, China. A multivariate regression model was derived to identify associations in physical activity levels between parents and children according to gender and weekday/weekends.

## METHODS

### Ethical approval

The study was carried out ethically and approved by the Ethical Committee of Shanghai University of Sport (No. 2014028).

### Participants

A sample of 346 parent–child triads were recruited from a larger measurement validation study of families with children attending four public and three private kindergartens in the Yangpu and Baoshan districts of Shanghai, China. Before subject recruitment, the principals and teachers of the kindergartens and parents were informed of the purpose and procedures of the study, which was approved by the Ethical Review Committee of the Shanghai University of Sport. The purpose of the study was explained to the father or mother of the participating families, who then gave written informed consent.

### Measures and procedures
#### Measurements of physical activity and sedentary behavior

Study participants wore an ActiGraph GT3X+ accelerometer (Actigraph, Pensacola, FL, USA) from 6 am to 11 pm every day for seven consecutive days from a Saturday through the following Sunday (five weekdays and one weekend) (*Fuemmeler, Anderson & Masse, 2011*; *Ridgers et al., 2014*). Participants were instructed to wear the accelerometer constantly except when bathing, swimming and sleeping. The accelerometer measures 4.6 cm × 3.3 cm × 1.5 cm, and it weighs 19 g. Its sampling frequency was set to 30 Hz, and the sampling interval (epoch) in the present study was set to be 1 s for children and 60 s for adults (*Ostbye et al., 2013*; *Pate et al., 2006*). Subjects wore their accelerometer on the waist, above the right hip, using an elastic belt (*Hesketh et al., 2014*). Accelerometer data were analyzed to measure the following parameters: daily duration of sedentary behavior (SB), light physical activity (LPA), moderate physical activity (MPA), and vigorous physical activity (VPA). The MVPA was the sum of MPA and VPA, while TPA was the sum of MVPA and LPA.

#### Demographics

Participants were asked to fill out three questionnaires. The children's questionnaire, which was filled out by the children's parents or guardians, asked about birth date, gender, daily care and early childhood education. Age in months was calculated as months from the birthdate until the measurement date. The parent's questionnaire asked about education level, monthly income, family structure, parent's working style and duration of daily contact with children (on weekdays and weekends). Finally, the Child Behavior Checklist (CBCL), which was filled out by the child's main preschool teacher, asked four items: "whether the child shows lack of concentration or non-persistent attention", "whether the child is introverted and unwilling to talk", "whether the child is over-fatigued" and "whether the child has slow actions or anenergia." Respondents could select a response of "not at all" (1 point), "occasionally" (2 points) or "frequently" (3 points). This test was used to assess the movement ability of preschool children, identify behavioral problems in participating children and ensure the validity of the collected data.
Weight and height were measured using standard physical fitness monitoring equipment. Body mass index (BMI) was calculated using the formula: BMI = (body weight in kg)/(height in m)$^2$. According to the International Obesity Task Force (IOTF), the children with BMI >25 were classified as overweight and >30 as obese (*Cole et al., 2000*).

### Fitness testing

Cardiorespiratory and motor fitness (*Esteban-Cornejo et al., 2014*) were assessed using a 20-m shuttle-run test, along with a 2 × 10-m shuttle-run test (*Leger et al., 1988*). Performance on each test was included in the regression analysis as a confounding factor. For the 2 × 10-m shuttle-run test, the preschool children were grouped into pairs, each of which was instructed to stand behind the starting line with legs apart, then to run immediately to the turn line after hearing the start signal, touch a car tire, then turn and run to the target line. Performance times were recorded to the nearest 0.1 sec, with another 0.1 sec added if the hundredths place was >0 (e.g., 0.13 was reported as 0.2).

For the 20-m shuttle-run test, the preschool children had to run back and forth for 20 m at an initial speed of 8.5 km/h, which increased by 0.5 km/h every minute in response to a whistle sound played on a CD (*Leger et al., 1988*). Maximal performance was determined when the child no longer kept pace or the child stopped because of exhaustion. Results were expressed in terms of stages, with one stage corresponding to approximately 20 m. A member of the study staff ran together with the children in order to avoid confusion.

## Data reduction

Duration of physical activity was estimated using a floating window algorithm (*Choi et al., 2011*). For accelerometer data to be considered valid, the accelerometer had to be worn for at least 8 h per day, and data had to be available for at least two weekdays and one weekend of the study period (*Ostbye et al., 2013*; *Ruiz et al., 2011*). SB, LPA, MPA and VPA measurements of children were categorized into the following groups (counts/min): <100, 100–1,680, 1,680–3,368, and ≥3,368 (*Pate et al., 2006*). The corresponding categories for parents were <100, 100–2,020, 2,020–5,999, and ≥5,999 (*Troiano et al., 2008*). Recordings of more than 20,000 counts/min were considered impossible and deleted (*Maher et al., 2014*; *Wang, Chen & Zhuang, 2013*). Data sampling and analysis parameters of this research, are all the often used and reasonable parameter assignment of existing index for measuring preschool children physical activity, so as to ensure the accuracy of the research results and comparability with other similar studies.

## Data analysis

All analyses were conducted using SPSS 22.0 (IBM, Chicago, IL, USA). A two-sided $p < 0.05$ was considered statistically significant. Results for normally distributed data were reported as mean ± standard deviation (SD), while results for skewed data were reported as median (interquartile range). Inter-gender differences were assessed for significance using the independent $t$ test for normally distributed data, the Mann–Whitney $U$ test for skewed data, or the chi-squared test for categorical data. Multiple linear regression was used to examine possible effects of parent's physical activity on preschool children's physical activity, after some factors such as age, BMI, family structure, family income, parents' daily

**Table 1 Characteristics of study subjects.**

| Characteristic | | Total ($n=247$) | Boys ($n=140$) | Girls ($n=107$) | $p$ |
|---|---|---|---|---|---|
| Age (month) | | $57.4 \pm 5.2$ | $57.9 \pm 5.2$ | $56.9 \pm 5.3$ | 0.927 |
| BMI (kg/m$^2$) | | $16.3 \pm 1.9$ | $16.6 \pm 1.9$ | $15.8 \pm 1.7$ | 0.083 |
| | Normal | 195 | 102 | 93 | |
| | Overweight/Obese | 52 | 38 | 14 | 0.011 |
| Child behavior score | Low (4–6 points) | 155 | 74 | 81 | – |
| | Median (7–9) | 82 | 56 | 26 | – |
| | High (10–12) | 10 | 10 | 0 | – |
| Cardiorespiratory fitness (laps) | | 11 (10–14) | 11 (9–14) | 12 (10–15) | 0.151 |
| Motor fitness (S) | | 7.0 (6.6–7.6) | 7.0 (6.5–7.6) | 7.1 (6.7–7.6) | 0.219 |
| Family structure | Living with both parents | 238 | 136 | 102 | – |
| | Other | 9 | 4 | 5 | – |
| Household income (RMB/month) | <4,000 | 6 | 4 | 2 | – |
| | 4,000–8,000 | 41 | 21 | 20 | – |
| | 8,001–15,000 | 108 | 63 | 46 | – |
| | 15,001–30,000 | 73 | 44 | 29 | – |
| | >30,000 | 18 | 8 | 10 | – |

Notes.
Values are reported as mean ± SD for normally distributed data, as median (interquartile range) for skewed data, or count for categorical data.
BMI, body mass index.

interaction and parent's working style were controlled. Then the effect of father's physical activity on preschool children's physical activity and the effect of mother's physical activity on preschool children's physical activity were examined respectively using linear regression model as above.

# RESULTS

Of the 346 parent–child triads initially recruited into the study, 99 were excluded due to inadequate data. As a result, 247 parent–child triads were included in the analysis, which comprised 86 fathers, 161 mothers, 140 boys and 107 girls (Table 1). The preschool children had a mean age of $57.5 \pm 5.2$ months. Male and female children did not differ significantly in age, cardiorespiratory fitness or motor fitness. However, BMI showed a tendency to vary with gender (boys, $16.6 \pm 1.9$; girls, $15.8 \pm 1.7$; $p=0.083$). A total of 38 boys and 14 girls were overweight or obese. The prevalence of these conditions differed significantly between genders ($p=0.011$; Table 1).

## Physical activity in preschool children and parents

Boys spent significantly longer amounts of time in sedentary behavior on weekdays ($596.9 \pm 68.8$ min/day) than on weekends ($537.5 \pm 89.6$ min/day; $p < 0.001$) and significantly less time in MVPA on weekdays ($73.3 \pm 18.4$ min/day) than on weekends ($77.8 \pm 26.2$ min/day; $p=0.013$). Similarly, boys' TPA was significantly lower on weekdays ($174.1 \pm 33.0$) than on weekends ($182.3 \pm 47.7$ min/day; $p=0.02$). Girls spent significantly longer amounts of time in sedentary behavior on weekdays ($604.8 \pm 71.2$ min/day) than on weekends ($531.7 \pm 83.9$ min/day; $p < 0.001$). Fathers and mothers alike spent significantly longer

**Table 2 Accelerometer-based physical activity (min/day) in preschool children and parents.**

| Parameter | | Total | Weekday | Weekend | p | Cohen's d |
|---|---|---|---|---|---|---|
| Boys' physical activity (n = 140) | Sedentary | 580.8 ± 61.4 | 596.9 ± 68.8 | 537.5 ± 89.6 | <0.001 | 0.74 |
| | MVPA | 74.6 ± 18.7 | 73.3 ± 18.4 | 77.8 ± 26.2 | 0.013 | −0.20 |
| | TPA | 176.6 ± 32.9 | 174.1 ± 33.0 | 182.3 ± 47.7 | 0.020 | −0.20 |
| Girls' physical activity (n = 107) | Sedentary | 585.7 ± 59.8 | 604.8 ± 71.2 | 531.7 ± 83.9 | <0.001 | 0.94 |
| | MVPA | 69.9 ± 15.0 | 69.8 ± 16.1 | 69.2 ± 18.8 | 0.729 | 0.03 |
| | TPA | 166.4 ± 27.0 | 165.8 ± 29.3 | 166.3 ± 35.9 | 0.878 | −0.02 |
| Fathers' physical activity (n = 86) | Sedentary | 422.8 ± 78.1 | 437.0 ± 94.4 | 394.1 ± 101.8 | 0.001 | 0.44 |
| | MVPA | 37.7 ± 22.7 | 39.5 ± 26.4 | 33.6 ± 23.1 | 0.026 | 0.24 |
| | TPA | 316.1 ± 68.2 | 318.2 ± 76.8 | 311.9 ± 78.7 | 0.477 | 0.08 |
| Mothers' physical activity (n = 161) | Sedentary | 399.5 ± 81.8 | 409.7 ± 93.2 | 369.7 ± 95.1 | <0.001 | 0.42 |
| | MVPA | 33.3 ± 21.3 | 36.1 ± 22.2 | 26.6 ± 27.5 | <0.001 | 0.24 |
| | TPA | 334.0 ± 81.6 | 335.3 ± 87.5 | 331.5 ± 94.6 | 0.552 | 0.04 |
| Parents' interaction time with children (h) | ≤1 | – | 61 | 32 | – | |
| | 2–4 | – | 137 | 19 | – | |
| | >5 | – | 49 | 196 | – | |

Notes.
Values are reported as mean ± SD for normally distributed data, as median (interquartile range) for skewed data, or count for categorical data.
MVPA, moderate-to-vigorous physical activity; TPA, total physical activity.

time in sedentary behavior on weekdays than on weekends (fathers, $p = 0.001$; mothers, $p < 0.001$); the same was observed with MVPA (fathers, $p = 0.026$; mothers, $p < 0.001$). Fathers and mothers showed similar TPA on weekends as on weekdays (Table 2).

Just over half (55.5%) of the fathers or mothers wearing an accelerometer spent 2–4 h with their children on weekdays (Table 2). One quarter (24.7%) of fathers or mothers spent less than 1 h with their children, while 19.8% spent more than 5 h. On weekends, 79.4% of fathers or mothers spent more than 5 h with the children, 13% spent less than 1 h, and 0.08% spent 2–4 h (Table 2).

## Correlation in sedentary behavior and activity level between parents and preschool children

There was a significant correlation on weekdays between parental sedentary activity and the sedentary activity of boys, girls and all children combined (all $p < 0.01$; Table 3). Fathers' sedentary levels correlated significantly with those of girls, but not those of boys; mothers' sedentary levels correlated significantly with those of boys, but not those of girls. Parents' and girls' MVPA significantly correlated with the MVPA of all children combined ($p < 0.01$ and 0.05, respectively), but not with boys' MVPA. Mothers' and girls' MVPA significantly correlated with each other ($p < 0.01$). Parental TPA significantly correlated with the TPA of boys ($p < 0.05$), girls ($p < 0.05$) and all children combined ($p < 0.01$). Mothers' TPA correlated with the TPA of girls or all children combined. However, no association was observed between fathers' and children's TPA levels.

On weekends, sedentary levels of parents correlated with those of girls and all children combined (all $p < 0.01$). Fathers' sedentary levels had more of an impact on girls' sedentary levels than mothers' levels did, while mothers' sedentary levels had more of an impact than

Table 3 Associations of parents' and preschool children's sedentary behavior with physical activity levels.

| Time | Subject | Sedentary behavior | | | MVPA | | | TPA | | |
|---|---|---|---|---|---|---|---|---|---|---|
| | | Boy | Girl | All | Boy | Girl | All | Boy | Girl | All |
| Weekday | Father | 0.244 | 0.318* | 0.279** | 0.032 | 0.332* | 0.153 | 0.181 | 0.134 | 0.153 |
| | Mother | 0.260* | 0.201 | 0.238** | 0.039 | 0.373** | 0.163* | 0.154 | 0.249* | 0.202* |
| | All | 0.278** | 0.255** | 0.270** | 0.038 | 0.331** | 0.155* | 0.168* | 0.205* | 0.191** |
| Weekend | Father | 0.050 | 0.453** | 0.235* | 0.339* | 0.444* | 0.329* | 0.279 | 0.337* | 0.272* |
| | Mother | 0.145 | 0.319** | 0.213** | 0.051 | 0.105 | 0.086 | 0.127 | 0.309* | 0.204* |
| | All | 0.109 | 0.357** | 0.212** | 0.123 | 0.201* | 0.145* | 0.177* | 0.314** | 0.231** |

Notes.

MVPA, Moderate-to-vigorous physical activity; TPA, total physical activity.

*$P < 0.05$.

**$P < 0.01$.

fathers' levels on the sedentary levels of all children together. However, no association was observed between parental and boys' sedentary levels.

Parents' MVPA levels correlated significantly with those of girls and all children combined (both $p < 0.05$). Fathers' MVPA levels correlated significantly with those of boys, girls and all children combined (all $p < 0.05$), and fathers' levels affected girls' levels more than boys' levels. However, no association was observed between mothers' MVPA and that of all children combined. Parental TPA levels correlated significantly with boys' ($p < 0.05$), girls' ($p < 0.01$) and all children's levels ($p < 0.01$). Both fathers' and mothers' TPA levels correlated with girls' and all children's levels (all $p < 0.05$), but boys' TPA levels showed no association with either fathers' or mothers' levels.

## Linear regression to identify correlations between parental and children's sedentary and activity levels

Linear regression models were developed to describe the effects of parental physical activity on children' physical activity on both weekdays and weekends (Table 4). Model 2 indicated that, after adjusting for age, BMI, family structure, household income and parent's daily interaction time with children and parent's working style, parental levels of sedentary behavior significantly correlated with preschool children's levels on weekdays ($p < 0.001$) and weekends ($p = 0.001$). The association was stronger on weekdays. On weekdays and weekends, mothers' sedentary behavior levels influenced preschool children's levels more so than fathers' sedentary behavior did. Parental MVPA levels showed significant associations with preschool children's levels on weekdays ($p = 0.018$) and weekends ($p = 0.029$), with a stronger association observed on weekdays. Mothers' MVPA levels correlated with preschool children's levels on weekdays, but not weekends. Conversely, fathers' MVPA levels were associated with preschool children's levels on weekends, but not weekdays. Parental TPA levels correlated significantly with preschool children's levels on weekdays ($p = 0.003$) and weekends ($p < 0.001$), with a stronger association observed on weekends. Mothers' TPA levels, but not fathers', correlated with preschool children's levels on weekdays. Fathers' and mothers' TPA levels were associated with preschool children's levels on weekends; fathers' levels exerted a stronger effect on children's levels than mothers' levels did.

**Table 4  Linear regression analysis to identify associations between parents' and preschool children's accelerometer-based physical activity.**

| | Sedentary behavior | | MVPA | | TPA | |
|---|---|---|---|---|---|---|
| | $\beta$ | $p$ | $\beta$ | $p$ | $\beta$ | $p$ |
| **Model 1[a]** | | | | | | |
| *Weekday* | | | | | | |
| Father ($n = 86$) | 0.279 | **0.009** | 0.153 | 0.159 | 0.153 | 0.160 |
| $R^2$ | | 0.067 | | 0.012 | | 0.012 |
| Mother ($n = 161$) | 0.243 | **0.002** | 0.162 | **0.041** | 0.207 | **0.009** |
| $R^2$ | | 0.053 | | 0.020 | | 0.037 |
| Total ($n = 247$) | 0.274 | **<0.001** | 0.154 | **0.016** | 0.194 | **0.002** |
| $R^2$ | | 0.072 | | 0.020 | | 0.034 |
| *Weekend* | | | | | | |
| Father ($n = 86$) | 0.235 | **0.029** | 0.329 | **0.002** | 0.272 | **0.011** |
| $R^2$ | | 0.044 | | 0.098 | | 0.063 |
| Mother ($n = 161$) | 0.214 | **0.006** | 0.086 | 0.287 | 0.206 | **0.009** |
| $R^2$ | | 0.040 | | 0.001 | | 0.036 |
| Total ($n = 247$) | 0.214 | **0.001** | 0.144 | **0.025** | 0.232 | **0.001** |
| $R^2$ | | 0.042 | | 0.017 | | 0.050 |
| **Model 2[b]** | | | | | | |
| *Weekday* | | | | | | |
| Father ($n = 86$) | 0.281 | **0.013** | 0.138 | 0.222 | 0.197 | 0.074 |
| $R^2$ | | 0.045 | | −0.002 | | 0.055 |
| Mother ($n = 161$) | 0.243 | **0.003** | 0.140 | **0.030** | 0.186 | **0.020** |
| $R^2$ | | 0.058 | | 0.057 | | 0.088 |
| Total ($n = 247$) | 0.276 | **<0.001** | 0.146 | **0.021** | 0.191 | **0.003** |
| $R^2$ | | 0.072 | | 0.040 | | 0.067 |
| *Weekend* | | | | | | |
| Father ($n = 86$) | 0.240 | **0.032** | 0.381 | **0.001** | 0.318 | **0.005** |
| $R^2$ | | 0.014 | | 0.055 | | 0.058 |
| Mother ($n = 161$) | 0.227 | **0.003** | 0.046 | 0.574 | 0.203 | **0.012** |
| $R^2$ | | 0.129 | | 0.037 | | 0.039 |
| Total ($n = 247$) | 0.211 | **<0.001** | 0.137 | **0.035** | 0.234 | **<0.001** |
| $R^2$ | | 0.082 | | 0.019 | | 0.041 |

**Notes.**

MVPA, moderate to vigorous physical activity;  TPA,  total physical activity.

The $p$ values less than 0.05 are bolded.

[a] Model 1: Unadjusted.

[b] Model 2: Adjusted for age, BMI, family structure, household income, child behavior score, parent–child interaction time per day and parent's working style.

## DISCUSSION

The present study showed a significant association between levels of sedentary behavior and physical activity of preschool children in Shanghai with the corresponding levels in their parents. Multivariate linear regression revealed significant differences in physical activity levels between weekdays and weekends, between fathers and mothers, and between boys and girls. This regression was adjusted for age, BMI, family structure, household income, parent–child interaction time per day and parent's working style.

### Parental and preschool children's sedentary behavior levels on weekdays and weekends

Levels of SB, MVPA, and TPA differed significantly between weekdays and weekends for parents and children. Sedentary behavior levels in boys, girls and parents were higher on weekdays than on weekends. This is consistent with the idea that on weekdays, most parents are at work and have little time to participate in planned physical activity. In contrast, preschool children stay in kindergarten between 7 am and 5 pm, with high sedentary behavior levels.

On weekdays and weekends, parents and preschool children spent more time in sedentary behavior than in physical activity. The factors behind this are likely similar to those reported in the study "Physical Activity and Health in Children and Adolescents" released by the Spanish government (*Merino & Briones, 2007*). These factors include: (1) overuse of electronic products such as television, computers, cell phones and tablets, which gradually replace time spent in outdoor activities; (2) car-based modern transportation, which reduces daily time spent walking; and (3) continuously accelerating urbanization. Parental sedentary behavior levels significantly affected preschool children's behavior, on weekdays and weekends (Table 3). Previous work has shown that children whose parents frequently engage in sedentary behavior (watching TV) also spend a substantial amount of time watching TV (*Jago et al., 2010*).

### Parental and preschool children's physical activity levels on weekdays and weekends

We found that parental physical activity had a significant influence on preschool children's physical activity ($p = 0.001$). These results are consistent with the finding that children of two active parents are 5.8 times more likely to be active than children of two inactive parents (*Oliver, Schofield & Schluter, 2010*). Meanwhile, we also found that the influence of parental physical activity differs between weekdays and weekends. Parents and preschool children showed lower MVPA and TPA levels on weekdays than on weekends. This suggests that on weekends, both parents and children have more opportunities to participate in physical activity and spend more time in leisure and entertainment activities instead of sedentary behavior. Consistent with this idea, most parents spent more than 5 h per day with their children on weekends, compared to 2–4 h per day on weekdays. In addition, parents' and children's MVPA correlated with each other, as did their TPA levels. It is likely that parents spend time with their preschool children in order to protect them (especially when activities take place outdoors), and they likely spend more time with them in the evenings. Through these interactions, parents can directly influence their children's physical activity (*Cools et al., 2011*).

### Analysis of discrepant influence of physical activity levels of parents on boys and girls

We examined to what extent the observed parental effects on children were dependent on gender. Our data indicated that parents' sedentary behavior and TPA were associated with the corresponding behaviors in their sons and daughters. The data also showed an association between parents' and daughters' MVPA levels. On weekends, parents' MVPA

as well as their sedentary behavior correlated with girls' levels, and parents' TPA levels correlated with those of boys. Previous work has reported higher physical activity levels in boys than in girls, especially in families where parents strongly support children's participation in physical activity (*Sterdt et al., 2014*). Data from 986 preschool children and 539 parents based on accelerometers, questionnaires and interviews indicated an association between parents' and daughters' sedentary time, but not between parents' and sons' sedentary time (*Jago et al., 2010*). Similarly, we found in the present study that physical activity levels were higher in boys than in girls, and parents' physical activity had a stronger influence on girls than boys. It may be that boys exhibit (or are encouraged to exhibit) greater autonomy, whereas girls tend to depend more strongly on the parents.

## Variation analysis of the influences of fathers and mothers on the physical activity level of preschool children

Our study also examined the relative influence of fathers' or mothers' physical activity on their children's activity. Most of the previous studies have focused on the mother-child link. For example, one study involving 554 preschool children aged 4 years old and their mothers showed associations between the two groups in accelerometer-measured sedentary behavior, LPA and MVPA (*Hesketh et al., 2014*). A cross-sectional study of 150 fathers of preschool children aged 3–5 years in which physical activity was assessed using the Pre-Physical Activity Questionnaire showed significant positive relationships between the two groups' physical activity on weekdays and weekends (*Vollmer et al., 2015*). Our study indicated that on weekdays, sedentary behavior of fathers was associated with that of their children, but the same was not observed for MVPA or TPA levels. In contrast, mothers' sedentary behavior, MVPA and TPA levels were associated with those of their children. On weekends, not only sedentary behavior but also MVPA and TPA levels of fathers were associated with those of their children, while sedentary behavior and TPA levels of mothers were associated with those of their children. These differences in maternal-paternal influence on children's physical activity likely reflect gender-based parenting roles within the traditional Chinese household. Our results suggest that on weekdays, the mother's physical activity influences that of the children to a greater extent than the father's does, while the converse is true on weekends. On weekdays, the father is more likely to focus on work and generating income, while the mother is more likely to look after the children and interact with them. Indeed, men in Chinese families typically spend less time in household activities than women (*Ng et al., 2014*). On weekends, in contrast, the father typically determines preschool children's physical activity. Future work may therefore need to focus on each parent separately and take into account whether physical activity occurs on a weekday or weekend; such work is needed to examine what characteristics of the father and mother influence their own levels of physical activity and their influence on their children's activity. For example, a mother's level of education and job type have been shown to affect preschool children's physical activity levels (*Ostbye et al., 2013*): women with more education and office work tend to elicit greater physical activity in their preschool children.

In summary, these findings suggest that preschool children's home environment must be taken into account when developing physical activity guidelines for Chinese preschool

children. These guidelines should inform and influence parent–child interactions. However, this research still belongs to a cross-sectional study, the results of the present study should be interpreted with caution because of several limitations. Since the accelerometer cannot measure all types of physical activity in preschool children, it is possible that TPA was underestimated. It is also possible that the physical activity in our study was affected by factors that we did not control, such as the weather and parents' and children's emotional states. We did not assess parental lifestyle, hobbies, interests or exercise skills, all of which can affect preschool children's physical and mental development, emotional state (*Lenze, Pautsch & Luby, 2011*), behavior acquisition (*Eisenstadt et al., 1993*), personality development and attitudes toward eating (*Brown & Ogden, 2004*)—all of which, in turn, can affect children's physical activity. We also did not take into account differences in the duration of daytime *vs.* nighttime interaction between parents and children on weekdays. Such work has been reported (*Fuemmeler, Anderson & Masse, 2011*; *Johansson et al., 2016*), but comparing that work to ours is difficult and potentially misleading because of substantial differences in the subjects and in mediating factors present.

Further studies are needed to systematically analyze associations between parents' and preschool children's physical activity over a given time period. Lastly, we did not take into account the possible influence of household composition such as the presence of grandparents and hired nannies. These factors are likely to influence preschool children's physical activity, particularly in the multigenerational, single-child households that still predominate in urban centers like Shanghai.

## CONCLUSION

The results of this study demonstrate that parental levels of physical activity and sedentary behavior affect the corresponding levels in their preschool children. These associations are significant and can be affected by many factors, include the parent's gender, the child's gender and whether it is a weekday or weekend. For example, maternal influence appears to be stronger during the weekdays, and paternal influence stronger on the weekends. Parents' activity levels influence girls' levels more strongly than they influence boys' levels.

### Funding
This work was supported by the grants from Shanghai Municipal Education Commission for promoting public health in students (No. HJTY-2014-A10) and the Ministry of Education of Humanities and Social Science Project (No. 17YJC890036) and the National Natural Science Foundation of China (No. 81703252). The funders had no role in study design, data collection and analysis, decision to publish, or preparation of the manuscript.

### Grant Disclosures
The following grant information was disclosed by the authors:
Shanghai Municipal Education Commission: HJTY-2014-A10.

Ministry of Education of Humanities and Social Science Project: 17YJC890036.
National Natural Science Foundation of China: 81703252.

## Competing Interests

The authors declare there are no competing interests.

## Author Contributions

- Chang Xu conceived and designed the experiments, prepared figures and/or tables, authored or reviewed drafts of the paper.
- Minghui Quan analyzed the data, performed the experiments, prepared figures and/or tables.
- Hanbin Zhang performed the experiments.
- Chenglin Zhou contributed reagents/materials/analysis tools, authored or reviewed drafts of the paper.
- PeiJie Chen conceived and designed the experiments, contributed reagents/materials/-analysis tools, authored or reviewed drafts of the paper.

## Human Ethics

The following information was supplied relating to ethical approvals (i.e., approving body and any reference numbers):

This study was approved by the Ethical Committee of Shanghai University of Sport (No. 2014028).

## Data Availability

The raw data has been provided as a Supplemental File.

## Supplemental Information

Supplemental information for this article can be found online at http://dx.doi.org/10.7717/peerj.4405#supplemental-information.

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
