# Peer review of "Impact of parents' physical activity on preschool children's physical activity: a cross-sectional study"

_PeerJ, doi:10.7717/peerj.4405_

## Round 0.1 · original submission · Major Revisions

I now have received two reviewers' comments. Although both reviewers expressed their interest in your study, several aspects of this manuscript should be revised to improve its clarity. Their observations are presented with clarity so I'll not risk confusing matters by belaboring or reiterating their comments. While I might quibble with the occasional point, I note that I regard the reviewers' opinions as substantive and well-informed. I believe that all of the highlighted reservations require contemplation and appropriate attention in revising the document if it is to contribute appropriately to Peerj and the extant literature. Please revise or refute according to the two reviewers' comments and provide a point by point reply in addition to the revised manuscript.

Tsung-Min Hung, Ph.D.
PeerJ editor
Distinguished professor
Department of Physical Education
National Taiwan Normal University

Reviewer 1 ·

Basic reporting

Using a regression analysis, the current study examined the associations of PA level (i.e., sedentary behavior, total physical activity, and moderate-to-vigorous physical activity) between parents and their pre-school children based on continuous time spanning five weekdays and two weekends, using an ActiGraph GT3X accelerometer. Results revealed a significant correlation between mothers’ and girls’ moderate-to-vigorous physical activity (MVPA) and total physical activity (TPA) on weekdays, while no such effect was observed between fathers’ and children’s TPA. In addition, Father’s MPVA was found to be correlated with those of boys’ and girls’, with paternal influence appearing to be stronger than maternal influence. The results indicate that parents’ PA may be associated with their preschool children’s physical activity level, with maternal influence stronger during the weekdays and paternal influence stronger on the weekends. Manuscript was written well with professional English throughout. Most of the literature references are provided sufficiently. However, the reviewer was expecting a more informative literature review on how the ActiGraph GT3X has been used and considered as a better tool compared to physical activity questionnaire and in-person observation. A paragraph to justify the measurement will be suggested. Specifically, after the brief description on the ActiGraph GT3X, on Line 93-94, the authors swiftly concluded their introduction and literature review without an adequate hypotheses.

Experimental design

no comment

Validity of the findings

no comment

Additional comments

The data collected from ActiGraph GT3X seems accurate and promising. However, the reviewer was curious if the working style from parents (i.e., sedentary versus non-sedentary working style) was entered as a controlled variable in the regression model. With the parents wearing the accelerator from 6 am to 11 pm every day, would the amount of physical activity measured simply reflect the nature of the parents' working style? Would a non-sedentary working style result in a more sedentary behavior after work for the parents? In addition, recording data from 6 am to 11 pm seems insufficient to the reviewer as parents might have diffident work time, while a 9 am to 5 pm time period for data collection may provide a more standard time frame for measuring physical activity and total activity amount. The reviewer was hoping that the authors could provide an explanation on how the time frame was determined in the current study.

Reviewer 2 ·

Basic reporting

Generally, this is an interesting paper and the ideas flow logically. But the introduction needs more detail and I suggest that you improve the description at lines 73-80 to provide more justification of the background information (adding the literature references).

Experimental design

The topic is timely and will be of interest to the readers of the journal. But the study design was not new and the statement of how this research filled the knowledge gap was not made clearly enough. I suggest that you provide more justification for the study design.

Validity of the findings

The authors did not justify whether the assumptions for linear regression was met before applying the model and this may result in Type I or Type II errors. In addition, although some confounders were controlled in the model, other factors like the environment of the neighborhood were not on the list. This may affect the conclusion of this research. Besides, the effect size was not reported.

Additional comments

Overall, the topic is timely and will be of interest to the readers of the journal. But numerous statements are made that reflect the authors’ opinion but they are portrayed as fact with no references cited or not sufficient evidence provided. I suggest that the authors improve the strength of the statistical analysis before making the final conclusion.

---

## Round 0.2 · accepted · Accept

I have now received the reviewer's comment that expressed satisfaction with your reply and revisions from previous comments. You and your coauthors have my congratulations. Thank you for choosing PeerJ as a venue for publishing your research work

Tsung-Min Hung, Ph.D.
PeerJ editor
Distinguished professor
Department of Physical Education
National Taiwan Normal University

Reviewer 1 ·

Basic reporting

The authors have successfully addressed the issued that the reviewer provided. The manuscript now is in a good shape and congratulate for the authors for the accomplishment!

Experimental design

no comment

Validity of the findings

no comment

Additional comments

no comment